# Identification of the Genes Involved in Anthocyanin Biosynthesis and Accumulation in *Taxus chinensis*

**DOI:** 10.3390/genes10120982

**Published:** 2019-11-28

**Authors:** Lisha Zhang, Xiaomei Sun, Iain W. Wilson, Fenjuan Shao, Deyou Qiu

**Affiliations:** 1State Key Laboratory of Tree Genetics and Breeding &Key Laboratory of Tree Breeding and Cultivation, National Forestry and Grassland Administration, The Research Institute of Forestry, Chinese Academy of Forestry, Beijing 100091, China; zlsxxb@aliyun.com (L.Z.); xmsun@caf.ac.cn (X.S.); qiudy@caf.ac.cn (D.Q.); 2CSIRO Agriculture and Food, P.O. Box 1600, Canberra, ACT 2601, Australia; Iain.Wilson@csiro.au

**Keywords:** anthocyanins, *Taxus chinensis*, gymnosperm

## Abstract

*Taxus chinensis* is a precious woody species with significant economic value. Anthocyanin as flavonoid derivatives plays a crucial role in plant biology and human health. However, the genes involved in anthocyanin biosynthesis have not been identified in *T. chinensis*. In this study, twenty-five genes involved in anthocyanin biosynthesis were identified, including chalcone synthase, chalcone isomerase, flavanone 3-hydroxylase, anthocyanidin synthase, flavonoid 3’-hydroxylase, flavonoid 3’,5’-hydroxylase, dihydroflavonol 4-reductase, anthocyanidin reductase, and leucoanthocyanidin reductase. The conserved domains and phylogenetic relationships of these genes were characterized. The expression levels of these genes in different tissues and different ages of xylem were investigated. Additionally, the anthocyanin accumulation in xylem of different ages of *T. chinensis* was measured. The results showed the anthocyanin accumulation was correlated with the expression levels of dihydroflavonol 4-reductase, anthocyanidin synthase, flavonoid 3’-hydroxylase, and flavonoid 3’,5’-hydroxylase. Our results provide a basis for studying the regulation of the biosynthetic pathway for anthocyanins and wood color formation in *T. chinensis*.

## 1. Introduction

Secondary metabolites are the products of plant adaptation to the environment during their evolution. Anthocyanins as a kind of secondary metabolite play a vital role in various processes of plant development, such as providing color to organs to attract pollinators, and of antioxidant activity to protect plants from injury by biotic and abiotic stress [1,2,3,4]. In addition, anthocyanidins are widely used in food, health care products, and drugs for their excellent antioxidant activity [5,6]. Anthocyanins can be divided into different types according to the different chemical groups on the lateral chains of the anthocyanidin skeleton. Typical anthocyanins have the anthocyanidin skeleton called 2-phenylbenzo-pyran that contains an aromatic benzene ring-oxygenated heterocycle-aromatic benzene ring (C6-C3-C6) structure [7,8]. Cyanidin, pelargonidin, and delphinidin are the three most fundamental anthocyanidins in plants as all other anthocyanidins are directly or indirectly modified from these three types. 

The structure and biosynthesis pathways of anthocyanins have been well studied. The genes related to anthocyanin biosynthesis have been reported in many plant species such as *Arabidopsis thaliana*, *Medicago sativa*, *Petunia hybrida*, and *Salvia miltiorrhiza* [9,10,11,12], which provide a good basis for understanding anthocyanin biosynthesis in these species. The biosynthesis of anthocyanin is complex. The 4-phosphoerythritose from the pentose phosphate pathway and the phosphoenolpyruvate from glycolysis are condensed into 7-phosphate heptanolose, which is modified into tyrosine and phenylalanine through a series involving the transformation of shikimic acid and the branching acid pathway (Figure 1). Tyrosine and phenylalanine can be catalyzed into p-coumaroyl-CoA through the phenylalkane metabolism pathway. P-coumaroyl-CoA and malonyl-CoA produced by glycolysis are the raw materials for the anthocyanins branch pathway. Chalcone synthase (CHS) is the key enzyme involved in anthocyanin biosynthesis; it catalyzes the condensation of P-coumaroyl-CoA and malonyl-CoA into chalcone. Chalcone is closed-loop for the formation of naringinin by chalcone isomerase (CHI). Naringinin is a key intermediate in the pathway of flavonoid biosynthesis. Flavanone 3-hydroxylase (F3H) catalyzes naringinint into dihydrokaempferol, which converts to leucopelargonidin under the catalysis of dihydroflavonol 4-reductase (DFR). Naringinin and dihydrokaempferol can also be catalyzed by flavonoid 3’-hydroxylase (F3’H) and flavonoid 3’,5’-hydroxylase (F3’5’H) into eriodictyol, pentahydroxyflavanone and dihydroquercetin, dihydromyricetin. Products of DFR belong to leucoanthocyanidin, which is catalyzed by anthocyanidin synthase (ANS) into cyaniding, pelargonidin, and delphinidin, the three most fundamental anthocyanidins. These three anthocyanidins generate various anthocyanins by oxidation, dehydration, and glycosylation. In addition, leucoanthocyanidin can also be catalyzed by anthocyanidin reductase (ANR) into flavane-3-alcohol, a precursor of procyanidin.

*T. chinensis* is a valuable woody species with important medicinal value. Paclitaxel, which is extracted from the bark of *T. chinensis*, is a natural antineoplastic drug that is widely used in the treatment of breast, ovarian, and lung cancers. In addition, the wood of Taxus species has many attributes, for example, good aesthetic appearance, purple red brown colored heartwood, straight texture, and rich elasticity [13,14]. The heartwood of *T. chinensis* contains high levels of anthocyanins and flavonoids. However, the genes involved in anthocyanins biosynthesis in *T. chinensis* have not yet been reported. To understand the correlation between gene expression and anthocyanin accumulation, we performed a systematic identification and analysis of genes related to anthocyanin biosynthesis in *T. chinensis*. In this study, 25 genes involved in anthocyanins biosynthesis in *T. chinensis* were identified through bioinformatics analysis and the conserved domains and phylogenetic relationships of these genes were analyzed. The expression profiles of these genes in different tissues and in different ages of *T. chinensis* xylem was also investigated. Moreover, the correlation between gene expression and anthocyanins accumulation was determined. Our results provide a basis for studying the regulation mechanism of biosynthetic pathway of anthocyanins in *T. chinensis*.

## 2. Materials and Methods 

### 2.1. Plant Materials 

*T. chinensis* was planted in the greenhouse located at Chinese Academy Forestry, Beijing, China. Leaf, xylem, phloem, root tissue, and one to four-year-old xylem were sampled from ten-year-old trees and stored in liquid nitrogen for RNA isolation. The bark was peeled from the developing stem, the phloem tissue was scraped from inside of the bark, and the xylem tissue was collected from the peeled log. Samples were taken from 3 plants and stored in ultra-cold storage freezer (−80 ℃) until use. One- to four-year-old xylem were weighed accurately for HPLC analysis of anthocyanin content.

### 2.2. Identification of Genes Related to Anthocyanin Biosynthesis

In order to predict the genes related to anthocyanin biosynthesis in *T. chinensis*, all the anthocyanin biosynthesis-related genes of Arabidopsis protein sequences were obtained from GenBank and were used as a query to search for homologous genes against the assembly of transcriptome (PRJNA580323) in *T. chinensis* using the tBLASTn algorithm with an e-value cut-off of 10^−10^. The predicted genes were inspected manually and corrected by sequence alignment with the genes involved in anthocyanin biosynthesis identified from other plant species by the BLASTx algorithm.

### 2.3. Gene Analysis in Bioinformatics and Phylogenetic Tree Construction

The molecular weight (Mw) and theoretical isoelectric point (pI) of deduced proteins involved in anthocyanin biosynthesis were analyzed by the Compute pI/MW tool on the ExPASy server (http://web.expasy.org/compute_pi/). Conserved domains were searched using CD-search tool on NCBI (https://www.ncbi.nlm.nih.gov/Structure/cdd/wrpsb.cgi) against the Pfam v31.0-16709 PSSMs database. Multiple amino acid sequence alignments were performed by Clustal Omega tool on EMBL-EBI (https://www.ebi.ac.uk/Tools/msa/clustalo). Phylogenetic trees were constructed by MEGA software (Philadelphia, CA, USA) with the neighbor joining (NJ) method (molecular evolutionary genetics analysis version 7.0) [15]. The protein sequences used for the sequence alignment and phylogenetic analysis were retrieved from NCBI. The accession numbers of protein sequences are used for the sequence alignment listed in Appendix A. 

### 2.4. Quantitative Real-Time Reverse Transcription-PCR (qRT-PCR)

Total RNA was isolated from the tissues of *T. chinensis* using the EASYspin plant total RNA isolation kit (Aidlab RN38, Beijing, China). RNA integrity and quantity were determined by 1.2% agarose gel electrophoresis and NanoDrop 1000C spectro-photometer (Thermo Scientific, Waltham, MA, USA). Reverse transcription was performed using the FastKing RT Kit (with gDNase) (TIANGEN, Beijing, China). The qRT-PCR are performed with SYBR^®^ rapid quantitative PCR Kit (KAPA KK4601, Pleasanton, CA, USA). Primers used for qRT-PCR are listed in Appendix A. *Tcactin* was used as a reference gene as described previously [16]. Gene expression levels were calculated according to the 2^-∆∆Ct^ method for the different tissues and aged xylem samples [17,18]. 

### 2.5. HPLC Analysis of Anthocyanin Contents

The conventional extraction method of anthocyanin is solvent extraction. Hydrochloric acid can prevent the degradation of non-acylation anthocyanin in the extraction process, and ultrasonic waves can improve the extraction efficiency [19]. Total anthocyanin was extracted by the ultrasonic assisted method [20]. Briefly, 200 mg of xylem was ground into powder in liquid nitrogen; the powder was then transferred into 2 mL methanol with 1% HCl at 4 °C protected from light for 12 h. After sufficient mixing, 2 mL of ddH_2_O and 4 mL chloroform were put in the solution and then centrifuged at 12,000 *g* for 20 min after ultrasonic treatment with an ultrasonic extractor for 1 h. The LC-20AD system (Shimadzu, Japan) and phenominex lunar C18 (4.6 µm × 250 mm) was used to analyze anthocyanin content according to previous studies [21,22]. The standard curve was established with cyanidin-3-glucoside chloride (Solarbio, Beijing, China) by the same method. Anthocyanin contents were calculated according to the standard curve.

## 3. Results

### 3.1. Identification of Genes Involved in Anthocyanin Biosynthesis in T. chinensis

In order to identify genes in the pathway of anthocyanin biosynthesis in *T. chinensis*, BLAST analysis of protein sequences of these genes in Arabidopsis against the assembly transcriptome data of the *T. chinensis* was performed using the tBLASTn algorithm [23]. A total of 25 genes were identified with 22 of these genes not previously identified. They belonged to nine families including CHS, CHI, F3’H, F3’5’H, F3H, LAR, DFR, ANR, and ANS. The identified genes are named *TcCHS*, *TcCHI1*-*TcCHI2*, *TcF3’H1*-*TcF3’H4*, *TcF3’5’H*, *TcF3H1*-*TcF3H5*, *TcLAR1*-*TcLAR2*, *TcDFR1*-*TcDFR8*, *TcANR*, and *TcANS* (Table 1). Sequence feature analysis of these genes included the length of ORFs (open reading frames), the molecular weight (Mw), the size of deduced proteins, and the theoretical pI which are listed in Table 1. All the deduced protein sequences contained the conserved domains (Appendix A), suggesting that they are the likely proteins involved in anthocyanin biosynthesis pathway in *T. chinensis*. Analysis of the deduced protein sequences against the Nr/nt database using BLAST algorithm showed that TcCHI2 has extremely high identities (98.58%) with TcCHI (AIQ85030.1, unpublished). TcF3’H4 and TcF3’5’H have 99.61% and 100% identity with CYP75B115 (ATG29929.1) and CYP75A77 (ATG29931.1) that were previously predicted from transcriptome data [24]. The other 22 genes were identified in this study. 

### 3.2. Phylogenetic Analysis and Expression Pattern of TcCHS

CHS plays a crucial role in anthocyanin biosynthesis pathway, and it is also one of the decisive factors affecting anthocyanin content [25]. CHS is one member of the type III polyketide synthase (PKS) family with two conserved domains (Chal_sti_synt_N and Chal_sti_synt_C). The Chal_sti synt_C domain is similar to domains of thiolase and beta-ketoacyl synthase. Only one CHS was identified in *T. chinensis*. The TcCHS contained two conserved motifs (Appendix A) and shared high sequence similarities with other plant species. Multiple protein sequence alignment of TcCHS and CHSs from other plant species indicated that all of them had the catalytic triad (Cys164-His303-Asn336) (Appendix A), which is the core active site of the type III PKS. In addition, the “gatekeepers” Phe-215 and Phe-265 and some inert active site residues such as Thr-132, Ser-133, Thr-194, Thr-197, Gly-256, and Ser-338 are highly conserved in the plants used in this study, except for Thr-197 of PcCHS which is replaced by Cys [26,27]. 

Amino acid sequences of CHS from 15 plant species including dicots, monocots, and gymnosperms were used to conduct the phylogenetic tree. The phylogenetic tree indicated that the CHSs were clustered into three clades (Figure 2a). TcCHS was included in clade1 with GbCHS, GmCHS, MdCHS, AtCHS, CsCHS, MsCHS, PtCHS, VvCHS, TuCHS, OsCHS, and GaCHS. Clade1 included both angiosperms and gymnosperms CHSs, indicating the function of these CHSs is highly conserved and consistent with a previous study [12]. The expression levels of TcCHS in roots, leaves, phloem, and xylem were analyzed using qPCR (Figure 2b). TcCHS exhibited the highest expression level in xylem, followed by phloem and leaves, with the lowest in roots. This indicated that TcCHS may play an important role in the development of xylem.

### 3.3. Phylogenetic Analysis and Expression Pattern of TcCHIs

CHI generally catalyzes naringenin chalcone into the flavanone (2S)-naringenin. It can be divided into four types including type I, type II, type III, and type IV. We identified two putative CHI genes from *T. chinensis*. The TcCHIs contained the conserved domain Chalcone_3 (pfam02431) (Appendix A). Multiple amino acid sequence alignments of TcCHIs and type I (AtCHI), type II (MsCHI), type III (AtFAP1), and type IV (AtCHIL) were performed [12,28,29] (Appendix A). Due to TcCHI1 possessing two Chalcone_3 domains, we separated it into two fragments for the alignment. The result showed that the two fragments of TcCHI1 share high sequence identity with AtCHI, indicating that TcCHI1 belongs to a type I CHI. Doubling of the functional domain might be caused by gene duplication. TcCHI2 has high sequence identity with AtCHIL, indicating it belongs to a type IV CHI. 

Amino acid sequences of CHI from plant species as previously used in CHS were used to conduct the phylogenetic tree except for *Ginkgo biloba* as no identified CHI from gymnosperms has been reported. The phylogenetic tree showed that the CHIs were divided into four clades (Figure 3a) and that most of the CHIs belong to type I, which is consistent with two previous studies [29,30]. TcCHI1 and TcCHI2 were clustered with AtCHI (type I CHI) and AtCHIL (type IV CHI), respectively, which is consistent with the multiple sequence alignment. There are some differences in the four types of CHI with regards to their biological functions [28]. Analysis of the qRT-PCR data showed that the two *TcCHI*s were expressed in all tissues of *T. chinensis* but exhibited different expression patterns. *TcCHI1* was highly expressed in leaves, followed by roots and xylem, and less in phloem, whereas *TcCHI2* showed the highest expression in xylem, followed by leaves and roots, and lower in phloem (Figure 3b). 

### 3.4. Phylogenetic Analysis and Expression Pattern of TcF3Hs and TcANS

F3H and ANS contain two conserved domains, including DIOX_N (pfam03171) and 2OG-FeII_Oxy (pfam14226) (Appendix A), and they belong to the 2-oxoglutarate dependent dioxygenase (2-ODD) superfamily. The DIOX_N domain is highly conserved in the N-terminal region with 2-oxoglutarate/Fe (II)-dependent dioxygenase activity [12,31,32,33]. F3H catalyzes naringenin, eriodictyol, and dihydrotricetin into dihydrokaempferol, dihydroquercetin, and dihydromyricetin. ANS catalyzes leucopelargonidin, leucocyanidin, and leucodelphinidin into pelargonidin, cyanidin, and delphinidin. We identified five putative F3H genes and one putative ANS gene from *T. chinensis*. The deduced amino acid sequences were aligned with F3Hs and ANSs of other plant species; the result showed that all of TcF3Hs and TcANS have the conserved H-x-D-xn-H motif (Appendix A) [34,35]. TcF3H1 and TcF3H2 have higher identity with AtF3H, whereas TcF3H3, TcF3H4, and TcF3H5 have higher identity with GbF3H. TcANS contains enzyme-specific active sites of ANS including Arg-298, which form the electrostatic interaction, and Phe-304, which binds the A-ring of the substrate and the “lip”structure (Val-235, Phe-334, IIe-338, and Leu-342) except for Val-235, which was replaced by Ile in *T. chinensis*.

The phylogenetic tree was constructed by using amino acid sequences of F3Hs and ANSs from plant species that are partially different from CHS, as there was no identified F3H and ANS from the same plant species. The phylogenetic trees showed that all the TcF3Hs and TcANS are clustered with gymnosperm F3Hs and ANSs (Figure 4a). TcF3H1 clustered with F3Hs from *Pinus radiata*, *Picea sitchiensis*, and *Picea glauca*, whereas TcF3H3, TcF3H4, and TcF3H5 were clustered with F3H from *Ginkgo biloba* (Figure 4a), implying possible different roles. The analysis of expression revealed that *TcF3H*s had different expression patterns in different tissues (Figure 4b). *TcF3H1* showed similar expression levels in each tissue, whereas *TcF3H3* and *TcF3H4* showed prominently high expression levels in roots and phloem, respectively. In addition, *TcF3H2* showed prominently low expression levels in xylem and roots. The expression level of *TcF3H5* increased slightly in leaves, phloem, xylem, and roots. These results indicate that different *TcF3H*s may have different functions in *T. chinensis*. *TcANS* exhibited high expression levels in phloem and xylem, implying that *TcANS* may be a candidate gene that participates in the color formation of red brown bark (Figure 4b).

### 3.5. Phylogenetic Analysis and Expression Pattern of TcF3’Hs and TcF3’ 5’H

F3’H and F3’5’H belong to the cytochrome P450-dependent monooxygenase superfamily and are haem-thiolate proteins involved in the oxidative degradation of various compounds [36,37]. They catalyze 3’ or 3’,5’ sites of the benzene ring in dihydrokaempferol oxygenated into taxifolin or dihydromyricetin. Four possible TcF3’Hs and one TcF3’5’H gene were identified in *T. chinensis*. All the deduced proteins contained p450 domain (pfam00067) (Appendix A), which belonged to the cytochrome P450 superfamily. The p450 domain usually contained conserved motifs including the proline-rich “hinge” region, oxygen binding pocket motif, E-R-R motif, and heme-binding domain [38,39]. Alignment of TcF3’Hs, TcF3’5’H and F3’Hs, F3’5’Hs from other plant species showed that the TcF3’Hs and TcF3’5’H contained the conserved motifs mentioned above (Appendix A). The proline-rich “hinge” region (PPGXXXP) is conserved at both ends, which is consistent with previous work [37]. The oxygen binding pocket motif (AGTDTSS) is almost completely conserved for F3’5’Hs but not for F3’Hs. It is replaced by “GGTESSA” for TcF3’H1, DcF3’H, and AcF3’H, while it is replaced by “AGTDTAS” and “GSTDTTS” for TcF3’H2 and TcF3’H3, respectively. Similar to the E-R-R motif and heme-binding domain, it is completely conserved for F3’5’Hs, whereas it has a one amino acid difference for F3’Hs. In addition, all of F3’5’Hs have the two conserved sequences “GHML” and “GLALQK” and F3’Hs do not. The differences in all these sequences may be responsible for the substrate specificity of the two enzymes [40]. 

Phylogenetic trees were constructed with F3’Hs and F3’5’Hs from plant species, of which F3’H and F3’5’H genes have been identified, including four TcF3’Hs and one TcF3’5’H (Figure 5a). TcF3’H1-TcF3’H2 clustered together with F3’H of *Pohlia nutans* (lower plant), whereas TcF3’H3-TcF3’H4 clustered together with F3’Hs of *Plectranthus scutellarioides* (dicot). TcF3’5’H clustered together with F3’5’Hs of *Glycine max* and *Salvia miltiorrhiza*. *TcF3’H1* showed higher expression level in roots than other tissues, while *TcF3’H2* showed the lowest expression level in roots (Figure 5b). *TcF3’H3* exhibited higher expression level in leaves and phloem than that in roots and xylem; *TcF3’H4* exhibited lower expression level in xylem than that in leaves, phloem, and roots. This possibly reflects the functional diversity of *TcF3’H*s in *T. chinensis*. *TcF3’5’H* exhibited lower expression level in leaves than that in phloem, xylem, and roots (Figure 5b). The different spatial and temporal expression patterns of *TcF3’H*s and *TcF3’ 5’H* indicate possibly different roles in the growth process of *T. chinensis*.

### 3.6. Phylogenetic Analysis and Expression Pattern of TcDFRs and TcANR

DFR catalyze dihydroflavonols into leucoanthocyanidins, which are the substrates of ANS. The products of ANS have two possible outcomes: one is conversion into anthocyanins by chemical modification, and the other is the production of flavane-3-alcohol catalyzed by ANR. DFR and ANR belong to the NAD/NADH-dependent epimerase family, which use nucleotide sugar as substrate for a variety of chemical reactions [41,42,43]. All of TcDFRs and TcANR contained the conserved domain epimerase (pfam01370) (Appendix A). DFR has three kinds of substrates including dihydrokaempferol, dihydroquerceti, and dihydromyricetin, which make contributions to different colors, in most plants, whereas DFR reduces the combining capability of substrates that cannot produce orange such as Petunia [44,45]. Multiple amino acid sequence alignments of GaDFR, MnDFR, GbDFR, VvDFR, AtDFR, and TcDFRs indicated that all TcDFRs have the enzyme specific active sites including Ser-128, Tyr-163, and Lys-167 (numbers refer to VvDFR) (Appendix A) [42]. There is a substrate specific recognition region between Ser-128 and Tyr-163 of which was reported that the conversion of Asn-133 to Asp-133 made the enzyme lose the combining capability of dihydrokaempferol as a substrate. TcDFR3 has Asn-133 in the substrate-specific recognition region alike that of GbDFR, VvDFR, and AtDFR, indicating that they can combine all the three substrates. *Morus notabilis* may lack the ability since MnDFR has Asp-133 in the recognition region. In addition, the other TcDFRs and GaDFR (*Gossypium arboreum*) have neither Asn nor Asp in the specific site, indicating that there are different substrate specific recognition mechanisms of DFR. Multiple amino acid sequence alignment of TcANR and ANRs from other species showed they were highly conserved. TcANR had active sites (Ser-130, Tyr-167, and Lys-171) and the NAD/NADH combination domain (G-G-X-G-X-X-A).

Amino acid sequences of DFR and ANR from plant species as previously used in CHS were used to conduct the phylogenetic tree, including eight TcDFRs and one TcANR, respectively (Figure 6a). It showed that TcDFR3 is closely related to GbDFR (*Ginkgo biloba*), which is consistent with the results of a previous study [45]. The other seven TcDFRs were closely related to angiosperm plants such as GaDFR (*Gossypium arboreum*), GmDFR (*Glycine max*), and NtDFR (*Nicotiana tabacum*) especially for TcDFR1, TcDFR2, and TcDFR5. TcANR clustered with PsANR (*Picea sitchensis*) and GbANR (*Ginkgo biloba*), which are gymnosperms. The analysis of expression profiles revealed that *TcDFR5* exhibited similar expression in the tissues analyzed; *TcDFR2* and *TcDFR6* exhibited prominently high expression level in leaves, *TcDFR3* and *TcDFR7* exhibited mainly high expression level in roots and *TcDFR1*, and *TcDFR4* exhibited high expression levels in phloem and xylem (Figure 6b). In addition, *TcDFR8* had almost no expression in leaves, phloem, and xylem, except for roots. The expression level of *TcANR* exhibited no significant difference among the different tissues (Figure 6b). Together with the expression data, the conservation and diversity of *TcDFR*s indicate their possible important role in anthocyanin biosynthesis in *T. chinensis*.

### 3.7. Phylogenetic Analysis and Expression Pattern of TcLARs

LAR is a NADPH-dependent enzyme that catalyzes leucoanthocyanidin into (+)-flavane-3-alcohol, which is the precursor of procyanidin [46]. LAR has been identified and characterized by the conserved domain NmrA (pfam05368) in a number of plant species such as *Vitis vinifera*, *Malus domestica*, and *Medicago truncatula*, and the crystal structure from *Vitis vinifera* has been analyzed [47,48,49,50]. Ser-118, Lys-140, Ile-162, and Asp-98 are residues associated with NADPH binding. In this study, we identified two TcLARs. Sequence feature of TcLARs is shown in Table 1. The deduced TcLAR proteins contain the NmrA domain (Appendix A) and Ser-118, Lys-140, and Ile-162 residues. It has been shown that Asp-98 is conserved in VvLAR, FtLAR, CsLAR, MrLAR, PaLAR, and PtLAR but is replaced by Asn, Glu, and Ser in CasLAR, TcLAR1, and TcLAR2 (Appendix A). In addition, Tyr-137 and His-122 are associated with OH-7 binding site; Met-136, Val-269, and Phe-272 are associated with A-C ring binding site; and Gly-93 and Met-136 are associated with the B-ring binding site. TcLARs all have the enzyme active residues except Met-136, which is replaced by Phe, similar to PaLAR and PtLAR. 

Phylogenetic trees were constructed with LARs from plant species of which LAR genes have been identified. The phylogenetic tree revealed that the LARs are divided into three clades: clade 1 (monocots), clade 2 (gymnosperms), and clade 3 (dicots) (Figure 7a). TcLARs and other gymnosperms LARs clustered together, which is consistent with the genetic classification of plants. The expression patterns of *TcLAR1* and *TcLAR2* are different in different tissues (Figure 7b). The expression levels of *TcLAR1* in leaves, xylem, and phloem were significantly higher than that in root, while the expression level of *TcLAR2* in roots was higher than that in other tissues, suggesting *TcLAR1* and *TcLAR2* may have different gene functions.

### 3.8. Correlation Analysis of the Expression of Genes Involved in Anthocyanin Biosynthesis and Anthocyanin Accumulation

In order to investigate the relationship between the expression levels of genes involved in anthocyanin biosynthesis and anthocyanin accumulation, the content of anthocyanin and the expression level of genes involved in anthocyanin biosynthesis of xylem ranging in age between one to four years in *T. chinensis* was measured by HPLC; 1% methanol hydrochloric acid was used as solvent in combination with ultrasonic extractor-assisted extraction at low temperature and dark conditions to obtain the best extraction. The results showed that one-year-old xylem had relatively higher anthocyanin content, which increased gradually from two-year-old to four-year-old xylem (Figure 8a). The chromatographic data is shown in Appendix A. The one-year-old xylem may contain more chlorophyll, covering up some red pigments that make the one-year-old branches appear green. 

Expression levels of genes involved in anthocyanin biosynthesis in *T. chinensis* were analyzed using qRT-PCR (Figure 8b,c). The results showed that *TcCHS* exhibited higher expression levels in one-year-old xylem than any other sample. *TcCHI1* and *TcCHI2* both exhibit higher expression levels in one-year-old xylem than others, especially for *TcCHI2*. This may suggest that the enhanced effect on anthocyanin biosynthesis in one-year-old xylem is also higher than that in older xylem. *TcF3H1*, *TcF3H2*, and *TcF3H3* exhibited higher expression levels in one-year-old xylem than older xylem; *TcF3H4* and *TcF3H5* exhibited similar expression level in the different ages of xylem, except for relatively high level of *TcF3H4* in four-year-old xylem. *TcF3’H2* and *TcF3’H4* exhibited higher expression levels in one-year-old xylem and three-year-old xylem, respectively, while *TcF3’H1* and TcF3’H3 exhibited higher expression levels in four-year-old xylem. *TcF3’5’H* exhibited higher expression level in one-year-old xylem than that in older xylem. *TcDFR*s exhibited similar expression patterns except for relatively high levels for *TcDFR1*, *TcDFR4*, and *TcDFR8* in one-year-old xylem. *TcANR* exhibited similar expression level in each tissue and higher levels in one-year-old xylem than other ages. *TcLAR1* showed similar expression, while *TcLAR2* showed the highest expression in one-year-old xylem. Taken together, the anthocyanin accumulation was correlated with the expression levels of *TcDFR1*, *TcANS*, *TcF3’H1*, and *TcF3’5’H*, suggesting these genes may play a vital role in the anthocyanin biosynthesis of xylem.

## 4. Discussion

Anthocyanins are an important type of secondary metabolites which play significant roles in colored plant tissues, such as flowers, fruits, and wood. Although the genes related to biosynthesis of anthocyanin have been identified in many plant species, they have not been systematically identified in *T. chinensis*. In this study, up to 25 genes putatively related to anthocyanin biosynthesis were identified in *T. chinensis*, including *TcCHS*, *TcCHI1*-*TcCHI2*, *TcF3H1*-*TcF3H5*, *TcF3’H1*-*TcF3’H4*, *TcF3’5’H*, *TcDFR1-TcDFR8*, *TcANS*, *TcANR*, and *TcLAR1*-*TcLAR2*. Only three of these have been previously reported. The sequence features and tissue-specific expression patterns of these genes in *T. chinensis* were analyzed. The conserved domains of deduced amino acid sequences of these genes and their phylogenetic relationships were performed and found to have high identity and close phylogenetic relationships with the corresponding genes from gymnosperms. 

*T. chinensis* is thought to have evolved in the Paleozoic era approximately 300 million years ago, surviving the harsh environment of the Quaternary glacier period. It belongs to relict plants of the Tertiary period and has a longer evolutionary period than angiosperms. Analysis of phylogenetic trees conducted in this study showed that most of the identified *T. chinensis* genes are located at the root of the phylogenetic tree, which is consistent with the evolutionary relationship between *T. chinensis* and other angiosperms [51,52]. However, different members in different gene families have different evolutionary relationships due to genetic differentiation [53]. Genes such as *TcF3’H1* and *TcLAR2* of which deduced proteins were closely related to angiosperms in phylogenetic trees may have more variation in their selection process. The gene family originated by gene duplication and divergence from a common ancestor. Interestingly, TcCHI1 possesses two Chalcone_3 domains in a tandem array in *T. chinensis*, whereas one CHI gene contains only one conserved domain similar to that found in other species. The sequence analysis of *Lotus japonicus* showed that there were four *CHI* genes in a tandem array in its genome [29]. Since *T. chinensis* precedes *Lotus japonicus* in the evolutionary relationship, we speculate that the ancestor gene of CHI family existed with several tandemly conserved domains and became a cluster of tandem genes through the long period of evolution. The number of genes identified in anthocyanin biosynthesis-related gene families in *T. chinensis* were different from other plant species such as *Arabidopsis thaliana, salvia miltiorrhiza*, and *Oryza sativa*, indicating that genes encoding the same enzyme may have different evolution patterns in different species (Appendix A) [12,28,54], for example, the number of *T. chinensis* DFR genes is significantly more than the number in *Arabidopsis thaliana, Salvia miltiorrhiza*, and *Oryza sativa*, suggesting that more gene duplication events occurred for DFR genes in *T. chinensis* (Appendix A).

The anthocyanin biosynthesis pathway has been extensively studied in plants. A number of studies about transcriptomics and metabonomics in anthocyanin biosynthesis revealed that the expression levels of structural genes play an important role in anthocyanin accumulation, especially *DFR, ANS*, and *F3H* [55,56,57,58]. *DFR* and *ANS* are considered as very important enzymes in anthocyanin biosynthesis, as they could direct the flavonoid flux into the anthocyanin branch [55,56,57,58]. Previous studies showed that the expression levels of *DFR, ANS,* and *F3H* and some transcription factors were positively correlated with anthocyanin accumulation. The expression levels of *TcDFR1, TcDFR4*, and *TcDFR5* showed higher expression levels in phloem than in other tissues. Similarly, the expression levels of *TcDFR3*and *TcDFR8* showed high expression in roots, possibility resulting in high anthocyanins accumulation in phloem and root. This is consistent with the red bark and red root of *T. chinensis.*

The wood of *T. chinensis* has high commercial value because of its aesthetic appearance, straight texture, high density, mechanical strength, rich elasticity, corrosion resistance, and purple red brown colored wood. The color of the wood is an important trait reflecting its quality and economic value. It has been shown that the formation of wood color is due to the existence of secondary metabolites in heartwood, including different kinds of phenolic compounds [59]. In order to study the effect of anthocyanin biosynthesis in wood formation, analysis of expression patterns of genes related to anthocyanin biosynthesis and the accumulation of anthocyanins in differently aged xylem of *T. chinensis* was conducted. Analysis of the expression levels of these genes in different tissues showed different expression patterns, suggesting potential different functions. The expression levels of *TcCHS*, *TcANS*, and *TcDFR1* showed relatively higher in xylem than in other tissues. The expression levels of *TcDFR1*, *TcANS*, *TcF3’H1*, and *TcF3’5’H* were observed to increase gradually from two-year-old to four-year-old xylem, especially for *TcDFR1* and *TcF3’5’H* that showed the highest expression level in one-year-old xylem, which is consistent with the trend of anthocyanin content accumulation (Figure 8a,b). In the flavonoid biosynthetic pathway, CHS, CHI, F3H, F3’H, and F3’5’H catalyze the flavonol biosynthesis, whereas DFR, ANS, and ANR lead to anthocyanin and procyanidins biosynthesis. F3’H and F3’5’H play primary roles in the diversification of anthocyanins through determining their B-ring hydroxylation pattern. DFR is specific for substrates which affect the anthocyanin composition and pigmentation [60]. ANS is a key enzyme for anthocyanin biosynthesis, which catalyzes the conversion of leucoanthocyanidin into anthocyanidin [61]. Our results indicate that the accumulation of xylem anthocyanins in *T. chinensis* has a close association with *TcDFR1*, *TcANS*, *TcF3’H1*, and *TcF3’5’H* their expression levels not only are consistent with the trend of anthocyanin content but also showed relatively high expression levels in xylem (Figure 8b,c), implying their potential important roles in anthocyanin accumulation in xylem, which provides a framework for future research focused on improved wood color. Another noteworthy result was that 14 of all the 25 genes exhibited the highest expression level in one-year-old xylem, which had the highest content of anthocyanins. This may be because one-year-old xylem as a young tissue has very high level of primary metabolism, which can provide the raw material for secondary metabolism. These results provide insights into the genes associated with wood color formation. Our results provide a basis for further research and possibly the manipulation of the regulation and accumulation of anthocyanins in *T. chinensis*.

## Figures and Tables

**Figure 1 genes-10-00982-f001:**
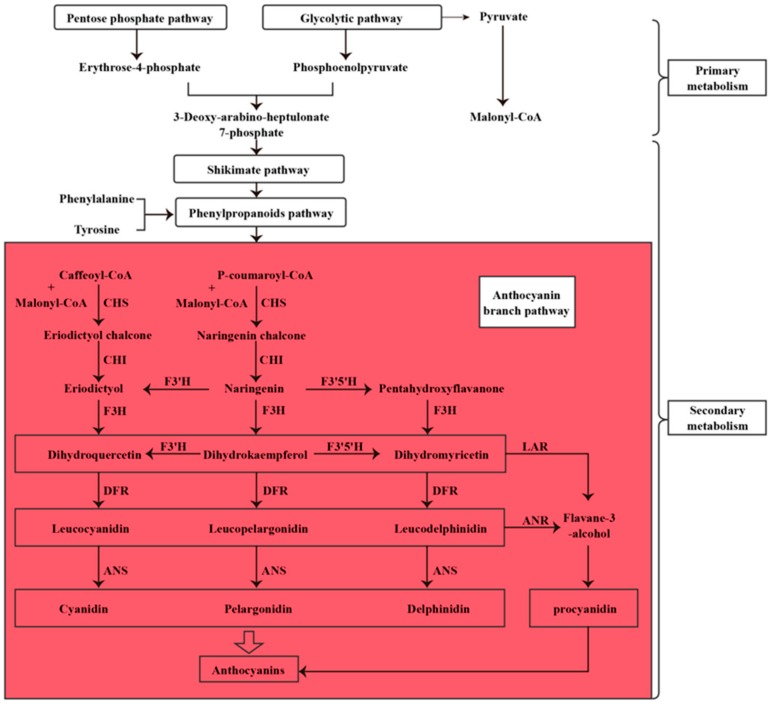
The biosynthetic pathways of anthocyanins. CHS, chalcone synthase; CHI, chalcone isomerase; F3H, flavanone 3-hydroxylase; F3’H, flavonoid 3’-hydroxylase; F3’5’H, flavonoid 3’,5’-hydroxylase; DFR, dihydroflavonol 4-reductase; ANS, anthocyanidin synthase; ANR, anthocyanidin reductase; LAR, leucoanthocyanidin reductase.

**Figure 2 genes-10-00982-f002:**
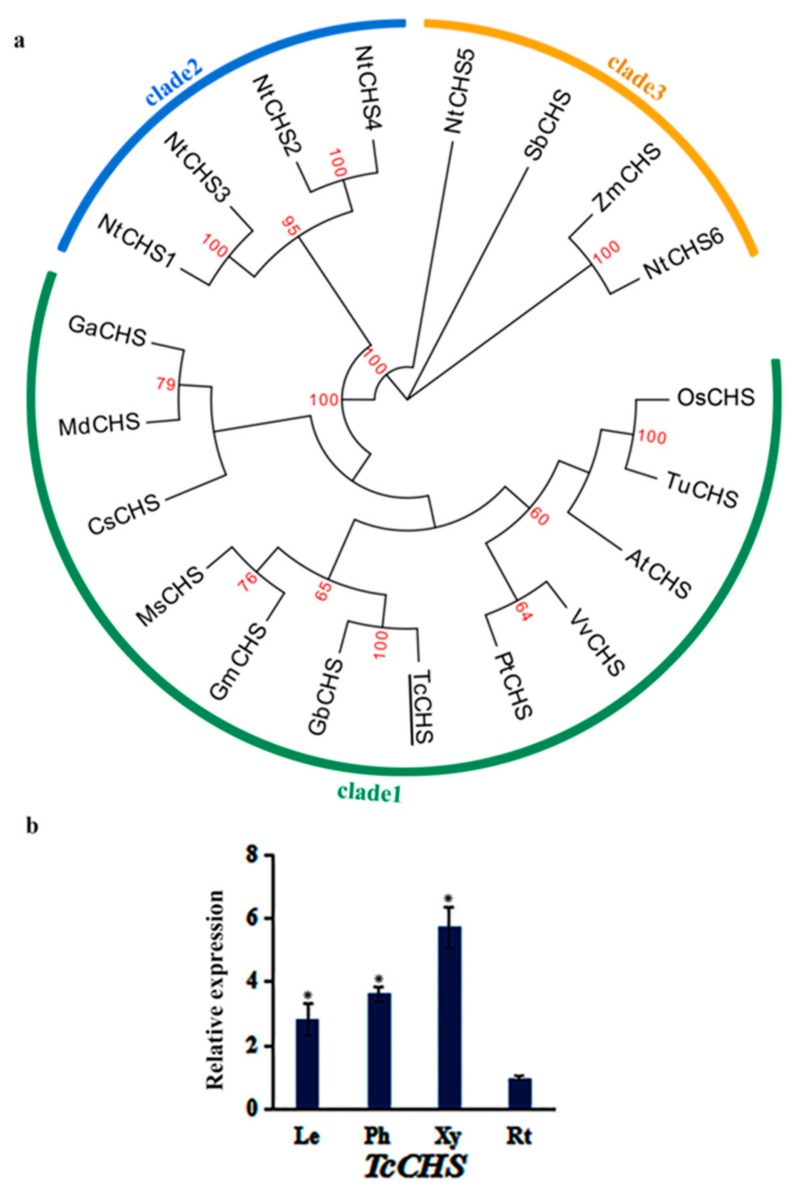
The phylogenetic relationship of TcCHS and expression patterns of *TcCHS*: (**a**) The phylogenetic relationship of CHSs in plants. CHS of *T. chinensis* was shown with underline. The sequences analyzed include *Arabidopsis thaliana* AtCHS (NP_196897.1); *Vitis vinifera* VvCHS (NP_001267879.1); *Glycine max* GmCHS (NP_001347353.1); *Populus trichocarpa* PtCHS (ABD24226.1); *Nicotiana tabacum* NtCHS1, NtCHS2, NtCHS3, NtCHS4, NtCHS5, and NtCHS6 (ANA78327.1, ANA78328.1, ANA78329.1, ANA78330.1, ANA78331.1, and ANA78438.1), *Camellia sinensis* CsCHS (AAT75302.1); *Medicago sativa* MsCHS (AAB41559.1); *Malus domestica* MdCHS (AGE84303.1); *Gossypium arboreum* GaCHS (KHG25969.1); *Zea mays* ZmCHS (NP_001149508.1); *Oryza sativa* OsCHS (BAA19186.2); *Triticum urartu* TuCHS (EMS66719.1); *Sorghum bicolor* SbCHS (XP_002441839.1); and *Ginkgo biloba* GbCHS (AAT68477.1). (**b**) The expression levels of *TcCHS* in leaves (Le), phloem (Ph), xylem (Xy), and roots (Rt): The expression level in roots was arbitrarily set to 1. Error bars represent the standard deviation of three technical PCR replicates. One-way ANOVA was calculated using IBM SPSS 19 software. *p* < 0.01 was considered statistically significant and was represented by asterisks.

**Figure 3 genes-10-00982-f003:**
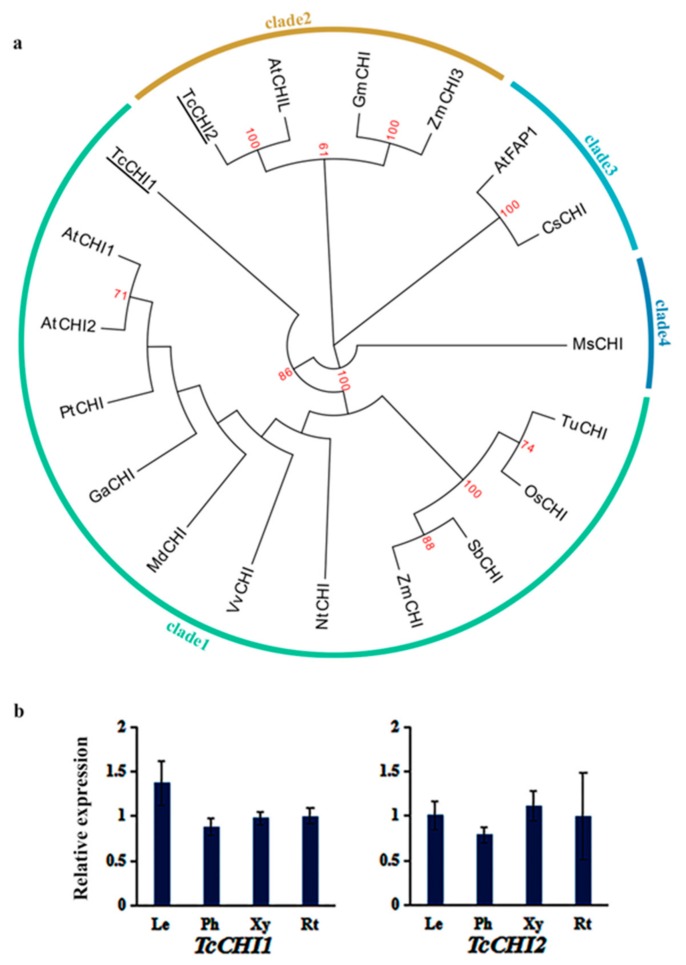
The phylogenetic relationship of TcCHIs and expression patterns of *TcCHI* genes: (**a**) The phylogenetic relationship of CHIs in plants. CHIs of *T. chinensis* were shown with underlines. The sequences analyzed include *Arabidopsis thaliana* AtCHI1, AtCHI2, AtCHIL, and AtFAP1 (NP_191072.1, NP_201423.2, NP_568154.1, and NP_567140.1); *Vitis vinifera* VvCHI (CAA53577.1); *Glycine max* GmCHI (NP_001351382.1); *Populus trichocarpa* PtCHI (XP_002315258.1); *Nicotiana tabacum* NtCHI (BAE48085.1); *Camellia sinensis* CsCHI (AGC30727.1); *Medicago sativa* MsCHI (AAB41524.1); *Malus domestica* MdCHI (XP_028956659.1); *Gossypium arboreum* GaCHI (KHG18033.1); *Zea mays* ZmCHI (NP_001144002.2); *Oryza sativa* OsCHI (AAO65886.1); *Triticum urartu* TuCHI (AHI94947.1); and *Sorghum bicolor* SbCHI (XP_002463631.1). (**b**) The expression levels of *TcCHIs* in leaves (Le), phloem (Ph), xylem (Xy), and roots (Rt): The expression level in roots was arbitrarily set to 1. Error bars represent the standard deviation of three technical PCR replicates. One-way ANOVA was calculated using IBM SPSS 19 software. *p* < 0.01 was considered statistically significant, and there was no significant difference between different tissues.

**Figure 4 genes-10-00982-f004:**
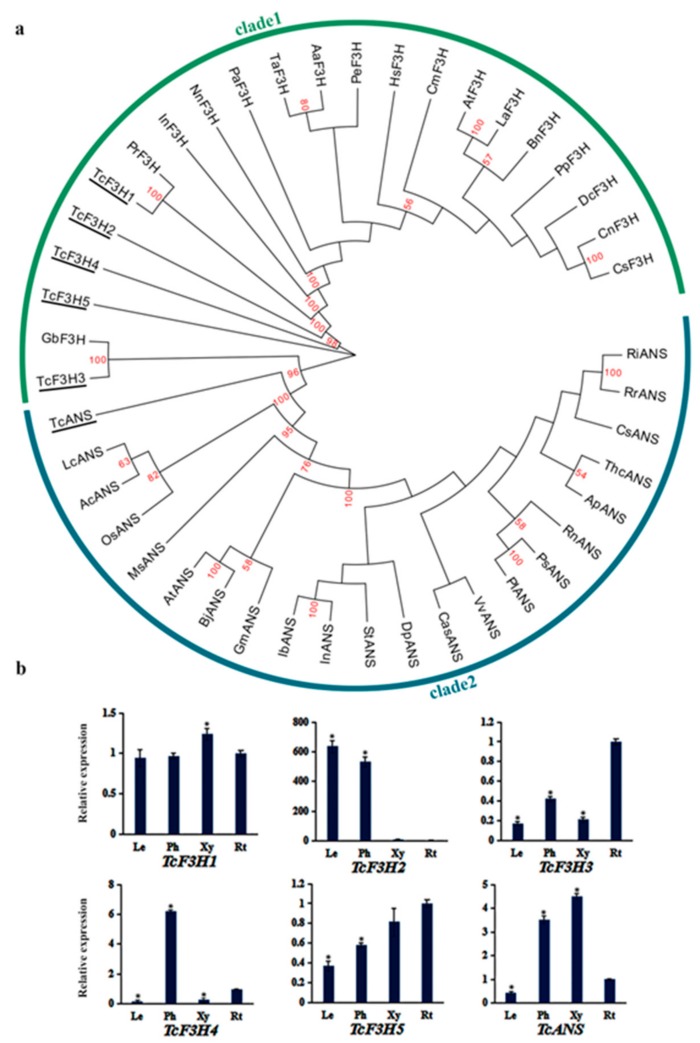
The phylogenetic relationship of TcF3Hs and TcANS and expression patterns of *TcF3H* and *TcANS* genes: (**a**). The phylogenetic relationship of F3Hs and ANSs in plants. F3Hs and ANS of *T. chinensis* were shown with underlines. The sequences analyzed include *Arabidopsis thaliana* AtF3H (NP_190692.1), *Boehmeria nivea* BnF3H (QBC98316.1), *Ipomoea nil* InF3H (BAA21897.1), *Clivia miniata* CmF3H (ARI70437.1), *Lepidium apetalum* LaF3H (ARA73611.1), *Triticum aestivum* TaF3H (ABR13013.1), *Daucus carota* DcF3H (AAD56577.1), *Anthurium andraeanum* AaF3H (ABI50233.1), *Phyllanthus emblica* PeF3H (AGT79807.1), *Nelumbo nucifera* NnF3H (AGT56413.1), *Pinus radiata* PrF3H (AGY80772.1), *Ginkgo biloba* GbF3H (ACY00393.1), *Camellia nitidissima* CnF3H (ADZ28514.1), *Camellia sinensis* CsF3H (AAT68774.1), *Persea americana* PaF3H (AAC97525.1), *Prunus persica* PpF3H (AQX36284.1), *Hololachna songarica* HsF3H (AEY81365.1), *Glycine max* GmANS (NP_001239794.1), *Vitis vinifera* VvANS (ABV82967.1), *Theobroma cacao* ThcANS (ADD51355.1), *Brassica juncea* BjANS (ACH58398.1), *Arabidopsis thaliana* AtANS (AEI99590.1), *Allium cepa* AcANS (ABM66367.1), *Acer palmatum* ApANS (AWN08246.1), *Citrus sinensis* CsANS (NP_001275784.1), *Rubus idaeus* RiANS (AQP31154.1), *Camellia sinensis* CasANS (ALF36156.1), *Rosa rugosa* RrANS (AKT74337.1), *Ribes nigrum* RnANS (AGI16383.1), *Dahlia pinnata* DpANS (AIZ70322.1), *Paeonia suffruticosa* PsANS (AIL29327.1), *Solanum tuberosum* StANS (NP_001274859.1), *Lycoris chinensis* LcANS (AGD99672.1), *Paeonia lactiflora* PlANS (AFI71900.1), *Ipomoea nil* InANS (BAB71811.1), *Oryza sativa* OsANS (CAA69252.1), *Ipomoea batatas* IbANS (ACT31916.1), and *Magnolia sprengeri* MsANS (AHU88620.1). (**b**) The expression levels of *TcF3Hs* and *TcANS* in leaves (Le), phloem (Ph), xylem (Xy), and roots (Rt): The expression level in roots was arbitrarily set to 1. Error bars represent the standard deviation of three technical PCR replicates. One-way ANOVA was calculated using IBM SPSS 19 software. *p* < 0.01 was considered statistically significant and was represented by asterisks.

**Figure 5 genes-10-00982-f005:**
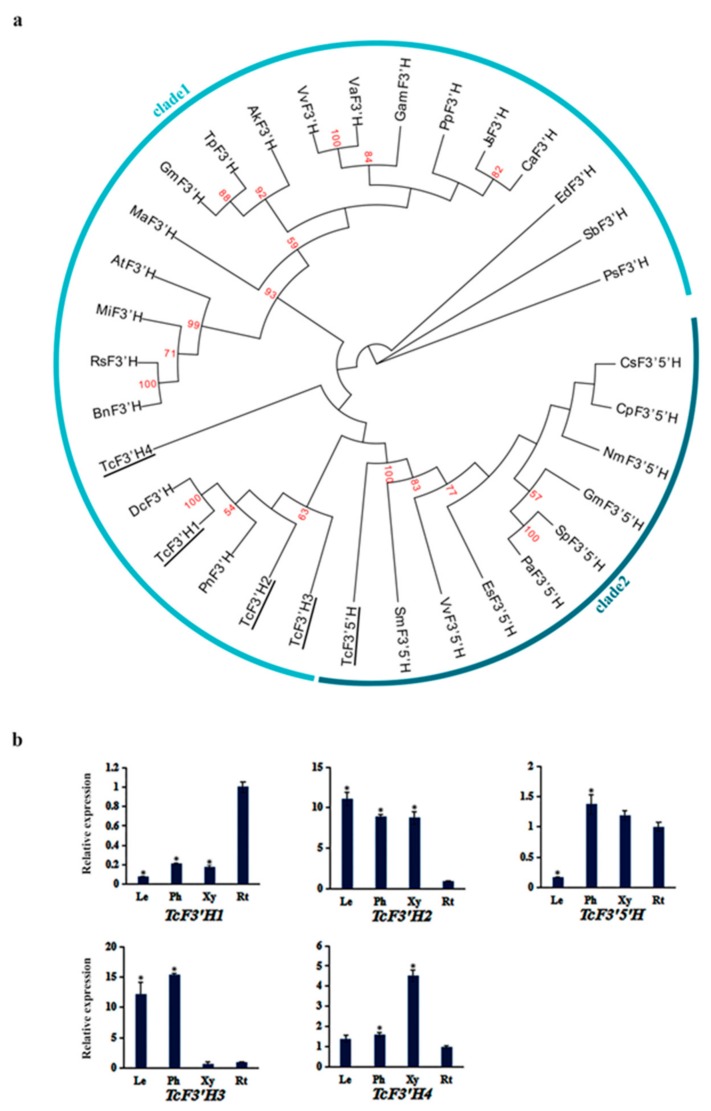
The phylogenetic relationship of TcF3’Hs and TcF3’5’H with F3’Hs and F3’5’Hs of other plant species and expression patterns of *TcF3’Hs* and *TcF3’5’H* genes: (**a**) The phylogenetic relationship of F3’Hs and F3’5’Hs in plants. F3’Hs and F3’5’H of *T. chinensis* were shown with underlines. The sequences analyzed include *Arabidopsis thaliana* AtF3’H (NP_196416.1), *Pyrus pyrifolia* PpF3’H (AWW17197.1), *Acacia koa* AkF3’H (JAI52338.1), *Juglans sigillata* JsF3’H (AYK27187.1), *Canarium album* CaF3’H (ATJ26448.1), *Morus alba* MaF3’H (AOV62762.1), *Garcinia mangostana* GmaF3’H (ACM62746.1), *Glycine max* GmF3’H (NP_001237015.1), *Vitis vinifera* VvF3’H (BAE47006.1), *Plectranthus scutellarioides* PsF3’H (APT37063.1), *Matthiola incana* MiF3’H (AAG49301.1), *Pohlia nutans* PnF3’H (AHI15955.1), *Sorghum bicolor* SbF3’H (AAV74195.1), *Brassica napus* BnF3’H (ABC58723.1), *Raphanus sativus* RsF3’H (BAX90121.1), *Trifolium pratense* TpF3’H (PNY13215.1), *Dracaena cambodiana* DcF3’H (AYM47547.1), *Egeria densa* EdF3’H (BAO56861.1), *Vitis amurensis* VaF3’H (ACN38268.1), *Glycine max* GmF3’5’H (NP_001236632.2), *Epimedium sagittatum* EsF3’5’H (ADE80942.1), *Cyclamen persicum* CpF3’5’H (ACX37698.1), *Solanum pennellii* SpF3’5’H (XP_015059023.1), *Vitis vinifera* VvF3’5’H (RVW36344.1), *Petunia axillaris* PaF3’5’H (AUI38393.1), *Salvia miltiorrhiza* SmF3’5’H (AWX67419.1), *Nemophila menziesii* NmF3’5’H (BBA68555.1), and *Camellia sinensis* CsF3’5’H (ASU87427.1). (**b**) The expression levels of *TcF3’Hs* and *TcF3’5’H* in leaves (Le), phloem (Ph), xylem (Xy), and roots (Rt): The expression level in roots was arbitrarily set to 1. Error bars represent the standard deviation of three technical PCR replicates. One-way ANOVA was calculated using IBM SPSS 19 software. *p* < 0.01 was considered statistically significant and was represented by asterisks.

**Figure 6 genes-10-00982-f006:**
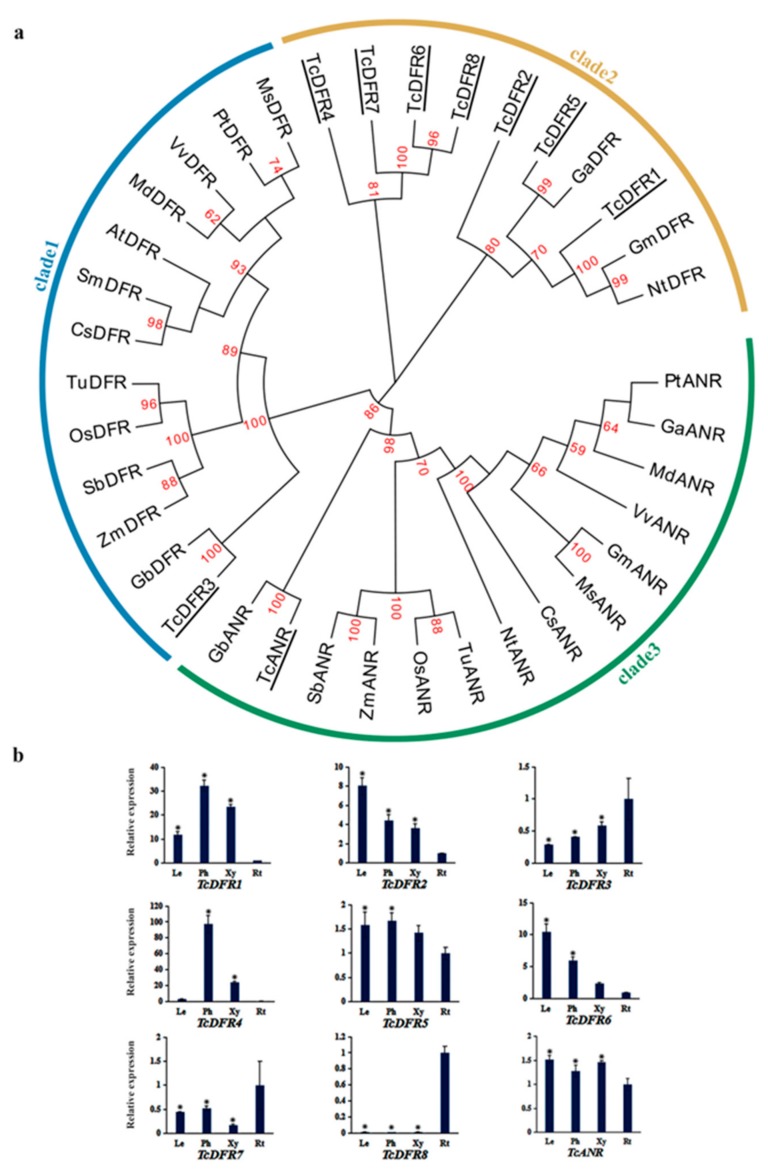
The phylogenetic relationship of TcDFRs and TcANR with DFRs and ANRs of other plant species and expression patterns of *TcDFRs* and *TcANR* genes: (**a**) The phylogenetic relationship of DFRs and ANRs in plants. DFRs and ANR of *T. chinensis* were shown with underlines. The sequences analyzed include *Arabidopsis thaliana* AtDFR (NP_199094.1), *Vitis vinifera* VvDFR (NP_001268144.1), *Glycine max* GmDFR (NP_001236658.1), *Populus trichocarpa* PtDFR (XP_006383711.2), *Nicotiana tabacum* NtDFR (AHZ08759.1), *Camellia sinensis* CsDFR (AAT66505.1), *Medicago sativa* MsDFR (AEI59122.1), *Malus domestica* MdDFR (AAD26204.1), *Gossypium arboreum* GaDFR (KHG24485.1), *Zea mays* ZmDFR (NP_001152467.2), *Oryza sativa* OsDFR (BAA36183.1), *Triticum urartu* TuDFR (EMS68193.1), *Sorghum bicolor* SbDFR (XP_002440593.1), *Ginkgo biloba* GbDFR (AGR34043.1), *Vitis vinifera* VvANR (BAD89742.1), *Glycine max* GmANR (NP_001241913.2), *Populus trichocarpa* PtANR (XP_002317270.2), *Nicotiana tabacum* NtANR (XP_016512400.1), *Camellia sinensis* CsANR (AHJ11240.1), *Medicago sativa* MsANR (ADK95116.1), *Malus domestica* MdANR (NP_001280930.1), *Gossypium arboreum* GaANR (NP_001316937.1), *Zea mays* ZmANR (ONM35828.1), *Oryza sativa* OsANR (XP_015637099.1), *Triticum urartu* TuANR (EMS67269.1), *Sorghum bicolor* SbAN R(XP_002447157.1), and *Ginkgo biloba* GbANR (AAU95082.1). (**b**) The expression levels of *TcDFRs* and *TcANR* in leaves (Le), phloem (Ph), xylem (Xy), and roots (Rt): The expression level in roots was arbitrarily set to 1. Error bars represent the standard deviation of three technical PCR replicates. One-way ANOVA was calculated using IBM SPSS 19 software. *p* < 0.01 was considered statistically significant and was represented by asterisks.

**Figure 7 genes-10-00982-f007:**
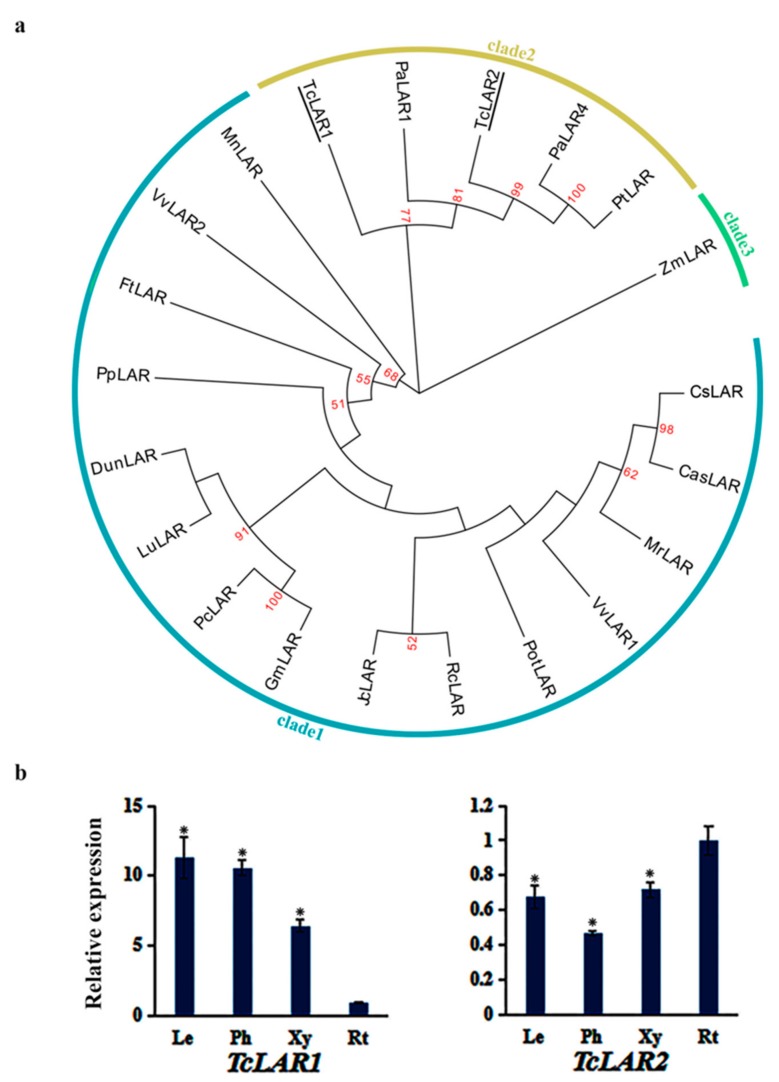
The phylogenetic relationship of TcLARs with LARs of other plant species and expression patterns of *TcLARs* genes: (**a**) The phylogenetic relationship of LARs in plants. LARs of *T. chinensis* were shown with underlines. The sequences analyzed include *Vitis vinifera* VvLAR1 (NP_001267887.1), *Vitis vinifera* VvLAR2 (NP_001268089.1), *Glycine max* GmLAR(NP_001352050.1), *Fagopyrum tataricum* FtLAR (AHA14498.1), *Zea mays* ZmLAR (NP_001148881.2), *Phaseolus coccineus* PcLAR (CAI56322.1), *Lotus uliginosus* LuLAR (AAU45392.1), *Desmodium uncinatum* DunLAR (CAD79341.1), *Ricinus communis* RcLAR (XP_002524404.2), *Picea abies* PaLAR1 (AHB89627.1), *Picea abies* PaLAR4 (AIA08662.1), *Morella rubra* MrLAR (AIX02997.1), *Pinus taeda* PtLAR (CAI56321.1), *Camellia sinensis* CsLAR (AZJ17294.1), *Populus trichocarpa* PotLAR (XP_024467009.1), *Morus notabilis* MnLAR (XP_010110804.1), *Camellia sinensis* CasLAR (ASU87431.1), *Jatropha curcas* JcLAR (XP_012082024.1), and *Prunus persica* PpLAR (XP_007222274.1). (**b**) The expression levels of *TcLARs* in leaves (Le), phloem (Ph), xylem (Xy), and roots (Rt): The expression level in roots was arbitrarily set to 1. Error bars represent the standard deviation of three technical PCR replicates. One-way ANOVA was calculated using IBM SPSS 19 software. *p* < 0.01 was considered statistically significant and was represented by asterisks.

**Figure 8 genes-10-00982-f008:**
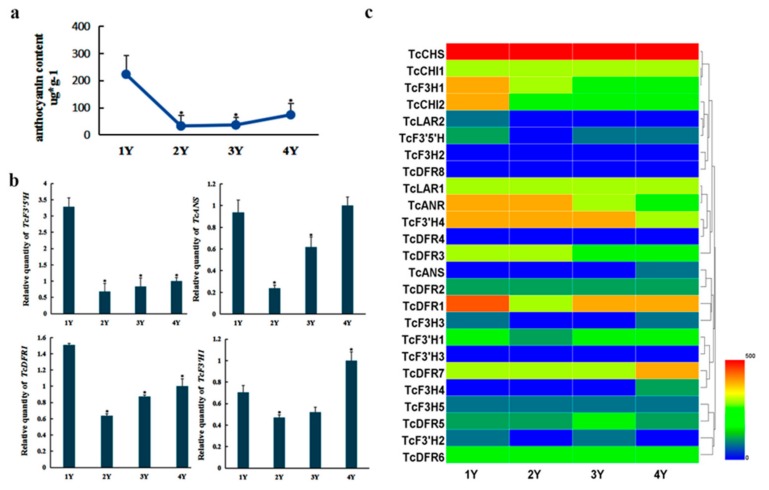
Accumulation of anthocyanins and expression levels of genes involved in anthocyanin biosynthesis in differently aged xylem in *T. chinensis*: (**a**) Accumulation of anthocyanins in differently aged xylem; 1Y, 2Y, 3Y, and 4Y represent one-year-old xylem, two-year-old xylem, three-year-old xylem, and four-year-old xylem, respectively. Error bars represent the standard deviation of three biological replicates. One-way ANOVA was calculated using IBM SPSS 19 software. *p* < 0.01 was considered statistically significant and was represented by asterisks. (**b**) Expression levels of *TcDFR1*, *TcANS, TcF3’H1*, and *TcF3’5’H* in differently aged xylem: The expression level in four-year old xylem was arbitrarily set to 1. Error bars represent the standard deviation of three technical PCR replicates. (**c**) Expression levels of genes involved in anthocyanin biosynthesis in different aged xylem. Different colors represent different expression levels. The expression level of each gene is compared to *Tcactin*×100.

**Table 1 genes-10-00982-t001:** Sequence features of genes related to anthocyanin biosynthesis in *Taxus chinensis.*

Gene Name	ORF (bp)	AA Len	Mw (Da)	pI
TcCHS	1191	396	43296.04	6.53
TcCHI1	1275	424	45693.46	5.26
TcCHI2	636	211	23169.55	5.08
TcF3H1	1083	360	40313.02	5.36
TcF3H2	1074	357	40289.22	6.09
TcF3H3	1020	339	37757.17	5.86
TcF3H4	1239	412	46216.48	5.79
TcF3H5	1110	369	41338.03	6.14
TcF3’H1	1518	505	57751.83	7.28
TcF3’H2	1521	506	57549.63	6.38
TcF3’H3	1518	505	56727.74	8.99
TcF3’H4	1551	516	57159.38	6.78
TcF3’5’H	1515	504	56269.67	9.33
TcDFR1	969	322	35761.2	5.63
TcDFR2	915	304	33774.41	5.6
TcDFR3	1053	350	38957.03	5.78
TcDFR4	1023	340	37318.85	5.49
TcDFR5	975	324	35997.33	6.33
TcDFR6	960	319	35470.71	5.95
TcDFR7	933	310	34393.59	5.44
TcDFR8	847	280	31009.74	5.83
TcANS	1050	349	39181.75	5.53
TcANR	1053	350	37627.39	7.6
TcLAR1	966	321	35852.81	5.35
TcLAR2	1215	404	44577.57	6.01

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
