# Peer review of "Identification of the Genes Involved in Anthocyanin Biosynthesis and Accumulation in Taxus chinensis"

_genes, 2019, doi:10.3390/genes10120982_

Round 1

Reviewer 1 Report

Author identified twenty five genes involved in anthocyanin biosynthesis. The expression levels of these genes in different tissues and different ages
of xylem were investigated. Moreover, the anthocyanin accumulation in xylem of different ages of T. chinensis was determined. The results showed the anthocyanin accumulation was correlated with the expression levels of dihydroflavonol 4-reductase, anthocyanidin synthase, flavonoid 3’-
hydroxylase and flavonoid 3’, 5’-hydroxylase.

The MS presented good molecular datas but lack of anthocyanin accumulation results. 

Although the overall interest and visibility of this work, some aspects should still be considered to improve the quality and objectiveness of this work. Overall, it is an important study, and should be considered for publication, once the issues have been resolved.

Major comments

Title should be changed. Because of author studied identification of the genes involved in anthocyanin biosynthesis as well as anthocyanin accumulation. Plant Materials sections provide detailed methods. HPLC analysis of anthocyanin contents: Author need to provide the HPLC data (chromatographic data should be provided). Discussion part need to be improved. The MS presented good molecular datas but lack of anthocyanin accumulation results. Gene expression related anthocyanin data should be provided. The expression levels of dihydroflavonol 4-reductase, anthocyanidin synthase, flavonoid 3’-
hydroxylase and flavonoid 3’, 5’-hydroxylase.

Author Response

Point-by-point responses to reviewer 1's comments

Comments and Suggestions for Authors

Author identified twenty five genes involved in anthocyanin biosynthesis. The expression levels of these genes in different tissues and different ages

of xylem were investigated. Moreover, the anthocyanin accumulation in xylem of different ages of T. chinensis was determined. The results showed the anthocyanin accumulation was correlated with the expression levels of dihydroflavonol 4-reductase, anthocyanidin synthase, flavonoid 3’-

hydroxylase and flavonoid 3’, 5’-hydroxylase. The MS presented good molecular datas but lack of anthocyanin accumulation results. Although the overall interest and visibility of this work, some aspects should still be considered to improve the quality and objectiveness of this work. Overall, it is an important study, and should be considered for publication, once the issues have been resolved.

Major comments

Title should be changed. Because of author studied identification of the genes involved in anthocyanin biosynthesis as well as anthocyanin accumulation.

The manuscript has been revised accordingly.

Plant Materials sections provide detailed methods.

The detailed method was added.

HPLC analysis of anthocyanin contents: Author need to provide the HPLC data (chromatographic data should be provided).

HPLC data has been provided in supplementary file.

Discussion part need to be improved. The MS presented good molecular datas but lack of anthocyanin accumulation results. Gene expression related anthocyanin data should be provided. The expression levels of dihydroflavonol 4-reductase, anthocyanidin synthase, flavonoid 3’-

hydroxylase and flavonoid 3’, 5’-hydroxylase.

Discussion was revised as advised and gene expression related anthocyanin data has been provided.

Reviewer 2 Report

The manuscript has been improved remarkably and the authors have followed most of my suggestions. However, there are still a couple of issues that should be addressed before publication:

1.- Line 499 (former 409). The authors said in the “response to reviewers” that this point was corrected but it was not. The sentence is still exactly the same in the new version. Again, I would like to point out that the fact that different paralogs have not overlapped expression patter not necessarily mean they had acquired different functions. This interpretation is not presenting all the relevant possibilities so I consider it incomplete and thereby incorrect. In my opinion, it should be removed from the discussion.

2.- The accession number provided by the authors regarding the transcriptomic data is not valid. I tried to search it within SRA database and I could not find it. The reason is that they used the PRJNAXXX number instead of the SRAXXX number. Please provide the SRA number in the final version of the manuscript.

Author Response

Point-by-point responses to reviewer 2's comments

Comments and Suggestions for Authors

The manuscript has been improved remarkably and the authors have followed most of my suggestions. However, there are still a couple of issues that should be addressed before publication:

1.- Line 499 (former 409). The authors said in the “response to reviewers” that this point was corrected but it was not. The sentence is still exactly the same in the new version. Again, I would like to point out that the fact that different paralogs have not overlapped expression patter not necessarily mean they had acquired different functions. This interpretation is not presenting all the relevant possibilities so I consider it incomplete and thereby incorrect. In my opinion, it should be removed from the discussion.

I'm very sorry for our mistake and the sentence has been removed.

2.- The accession number provided by the authors regarding the transcriptomic data is not valid. I tried to search it within SRA database and I could not find it. The reason is that they used the PRJNAXXX number instead of the SRAXXX number. Please provide the SRA number in the final version of the manuscript.

The transcriptomic data was uploaed to SRA on November 2, 2019 and we received an Email by the NCBI SRA Submissions Staff who said “Please reference PRJNA580323 in your publication. This BioProject accession number is provided above in lieu of SRP and should be used in your publication as it will allow better searching in Entrez. Your SRA records will be accessible with the following link after the indicated release date: https://www.ncbi.nlm.nih.gov/sra/PRJNA580323.”

The data will be released immediately after our paper published.

Round 2

Reviewer 1 Report

Requested corrections were done by authors. 

This manuscript is a resubmission of an earlier submission. The following is a list of the peer review reports and author responses from that submission.

Round 1

Reviewer 1 Report

The wood of Taxus chinensis seems to have an important commercial value not only because of the mechanical properties but also because of the colour. This relevant trait arises from the accumulation of secondary metabolites in the wood such as anthocyanins. In this work, Zhang and co-workers have performed a systematic search for anthocyanin biosynthesis orthologues in Taxus chinensis and an expression pattern of those genes in some tissues of interest. Although the current version of the manuscript is largely descriptive, the research is technically sound, and provide some interesting new data for those working in this field. However, authors should deal with a number of important issues prior to publication:

Major points:

Article title should be changed. I suggest not including the “…and their expression correlation with anthocyanin accumulation…” part. I think this way the title will be more clear for readers. In this work, authors have performed a search for orthologues using a transcriptomic data. If this data does not belong to authors, the source of information must be cited. If this data belongs to authors, it must be uploaded to a repository and a reference number must be provided to make it accessible for everybody. Names of the different sections: I suggest to avoid using the word “characterization” as authors do not provide any data beyond in silico and expression analysis. Instead I suggest to name these sections as “Phylogenetic analysis and expression pattern of ___”. To be able to easily compare among different species, I think authors must include the same species in all the phylogenetic analyses. If for a specific protein type, an additional species is analysed, it must be explain. Besides, the chosen species must be explain and the abbreviations must be detailed in every single figure legend. Figures 2b, 3b, 4b, 5b, 6b, 7b and 8a. A Y-axis must be included. It is necessary to include a statistical analysis to be able to differentiate between significantly different values and those that are not. Line 398-399. This sentence from the discussion highlights the need of a new table where authors compare the number of genes for every enzymatic activity in all the species analysed. Line 409. I do not agree that different expression patterns among paralogs indicate different functions. It may indicate a divergent evolution of their promoters providing tissue-specialization.

Minor points:

As I put in the form, the manuscript requires an extensive English editing. Some selected examples of this are found in lines 53,82, 180, 241, 276, 419. I consider that Figure 1 could be improved. Authors can make it wider, choose another colour and provide a less blurred file for the final version. Line 82. I do not consider that the results presented provide any insight of the regulatory mechanisms behind this pathway. Consider remove this sentence. Line 90. In general, the 2.1 section could be better written but I do not think that authors can consider freezing samples as a “treatment”. Line 95. 10-10 instead of 10-10 Line 99. Genes do not have molecular weight of isoelectric point. Those parameters a for proteins. Line 116. Authors should clarify why they use different calculation systems depending on the experiment. If possible, consider perform all of them in the same way. Line 122. I guess authors forgot to mention that they used liquid nitrogen to grind the tissue. Line 125. What equipment was used for the ultrasonic treatment? Line 147. All the abbreviations used in the upper part of the table must be explain in the bottom part (Mw, pI, AA, ORF…). Line 192. The sentence “There are some differences in the four types of CHI with regards to their biological functions.” In my opinion does not provide anything relevant. Consider expand this information or remove it. In the next sentence it is said that chinensis contained two types of CHI and according to authors it suggest a greater ability to synthesize flavonoids. Is this supported by anything else? Based exclusively on the gene copy number I do not think authors can go that further. Line 202. Similar to what authors do in the line 232, here they should cite relevant literature about these enzymes. I consider this specially relevant for 2OGDs, which lately are attracting a lot of attention in the research (Kawai et al 2014 Plant Journal; Mateo-Bonmatí et al. 2018 Plant Cell; Nadi et al. 2018, Molecular Plant, among others). Line 258. Please explain what “complex evolution” means here. Line 354. I do not understand what “relative to Tcactin*100” means here. Please clarify. Several spelling mistakes. Lines 105, 113, 202, 216, 217, 307, 329, 330, 349 and 410.

Reviewer 2 Report

The author studied twenty five genes such as chalcone synthase, chalcone isomerase, flavanone 3-hydroxylase, anthocyanidin synthase, flavonoid 3’-hydroxylase, flavonoid 3’, 5’-hydroxylase, dihydroflavonol 4-reductase, anthocyanidin reductase and leucoanthocyanidin reductase involved in anthocyanins biosynthesis of Taxus chinensis. The results showed the anthocyanin accumulation was correlated with the expression levels of dihydroflavonol 4-reductase, anthocyanidin synthase, flavonoid 3’-hydroxylase and flavonoid 3’,5’-hydroxylase in T. chinensis xylem. The biosynthetic pathway for anthocyanins and wood color formation in T. chinensis. Although the overall interest and visibility of this work, some aspects should still be considered to improve the quality and objectiveness of this work. Overall, it is an important study, and should be considered for publication, once the issues have been resolved.

If possible, try to add anthocyanin data using HPLC.
Overall, this manuscript needs more discussion about experimental results. Please speculate about the reasons to the obtained results. Interpretation is not enough. Need to discuss in details. Need to provide recent references.